# Effect of Atomoxetine on Behavioral Difficulties and Growth Development of Primary School Children with Attention-Deficit/Hyperactivity Disorder: A Prospective Study

**DOI:** 10.3390/children9020212

**Published:** 2022-02-06

**Authors:** Huiya Mei, Ruijin Xie, Tianxiao Li, Zongxin Chen, Yueying Liu, Chenyu Sun

**Affiliations:** 1Department of Pediatrics, Affiliated Hospital of Jiangnan University, No. 1000, Hefeng Avenue, Wuxi 214122, China; 6212809027@stu.jiangnan.edu.cn (H.M.); 6202809028@stu.jiangnan.edu.cn (R.X.); 6192807008@stu.jiangnan.edu.cn (T.L.); 2The First Affiliated Hospital of Soochow University, No. 188, Shixin Avenue, Suzhou 215000, China; sdfyyczx@163.com; 3AMITA Health Saint Joseph Hospital Chicago, 2900 N. Lake Shore Drive, Chicago, IL 60657, USA

**Keywords:** ADHD, atomoxetine, insulin-like growth factor 1, behavioral difficulties, growth development

## Abstract

(1) Objective: Atomoxetine is a selective norepinephrine reuptake inhibitor used to treat attention-deficit/hyperactivity disorder (ADHD) in children over six years old. Although it is common knowledge that primary school children with ADHD often present with difficulties in the morning prior to school and in the evening, these two periods, and the family interactions they involve, are often neglected in studies of ADHD. Questionnaire–Children with Difficulties (QCD) has been widely used in China to evaluate parents’ perceptions of ADHD and patients’ daily behaviors during different times. In the long term, the efficacy and safety of atomoxetine have been well established in previous studies. Still, the short-term effects of atomoxetine treatment on serum growth parameters, such as IGF-1, IGFBP-3, and thyroid function, are not well documented. Therefore, this study was the first one using the QCD to quantify the efficacy of atomoxetine treatment in the morning prior to school and in the evening, and has investigated the possible influence on the growth parameters of Chinese primary school children with ADHD. (2) Method: This prospective study was conducted at the Department of Pediatrics at the Affiliated Hospital of Jiangnan University from August 2019 to February 2021. Changes in the children’s behavior and core ADHD symptoms following treatment were assessed using three parent-reported questionnaires, including Children with Difficulties (QCD), the Swanson, Nolan, and Pelham IV scale (SNAP-IV), and the Conners’ parents rating scales (CPRS). The height, weight, and body mass index (BMI) were measured and corrected to reflect the standard deviations (SDS) in Chinese children based on age and gender. Serum growth parameters, such as insulin-like growth factor 1 (IGF-1), insulin-like growth factor-binding protein 3 (IGFBP-3), and thyroid function, were also measured to assess the children’s growth development. Any adverse drug reactions were assessed every three weeks. (3) Result: Finally, 149 children were enrolled in this study, and they completed 12 weeks of atomoxetine treatment. The QCD results indicated that the atomoxetine treatment could significantly alleviate behavioral difficulties in primary children with ADHD, especially in the morning prior to school (*p* < 0.001, r = 0.66) and in the evening (*p* < 0.001, r = 0.73). A statically significant decrease in weight SDS (*p* < 0.05) was noted during treatment, but the effect size was slight (r = 0.09). The atomoxetine treatment had no significant impact on height SDS, BMI SDS, and serum growth parameters, such as the levels of IGF-1, IGFBP-3, and thyroid function. The SNAP-IV results showed a significant improvement in the core symptoms of ADHD, while the CPRS results indicated a significant improvement in controlling ADHD symptoms across two different domains, learning problems (r = 0.81) and hyperactivity (r = 0.86). No severe adverse reactions were observed in the course of treatment, and the most common adverse reactions were gastrointestinal symptoms. (4) Conclusions: Atomoxetine is an effective and safe treatment for primary school children with ADHD. In China, it may be an excellent choice to alleviate parenting stress and improve the condition of primary school children with ADHD. Moreover, our study indicated that the serum levels of IGF-1 and IGFBP-3 were within the normal range in newly diagnosed ADHD children, and atomoxetine will not affect the serum concentration of growth parameters, such as IGF-1, IGFBP-3, and thyroid function, in the short term. However, the treatment may reduce appetite, resulting in a reduction in the Children’s weight for a short period. Further observational studies to monitor the long-term effects of atomoxetine on primary school children are recommended.

## 1. Introduction

Attention-deficit/hyperactivity disorder (ADHD) is the most common childhood neurodevelopmental disorder, with a prevalence of approximately 5% worldwide [1]. It is defined as a persistent pattern of inattention, hyperactivity, and impulsivity that significantly impedes children’s academic and social development [1,2,3]. If left untreated, ADHD may cause long-term morbidity and poor quality of life in adulthood [2]. Therefore, school-aged children with ADHD must receive essential medical treatment.

Behavioral and psychosocial treatments (BPT) are the recommended nonpharmacological treatments to alleviate core the symptoms of ADHD, and they are the first-line treatments for very young children (<5 years old) [4,5]. However, the efficiency of BPT remains questionable, with controversial evidence and conclusions in previous studies [5]. Pharmacological treatment is still widely used as ADHD treatment in most clinical practices and guidelines [6,7,8]. The current standard pharmacological treatment for primary school children with ADHD involves using stimulants or non-stimulants that regulate catecholamines [3]. In China, methylphenidate is widely used as a central nervous system stimulant, while atomoxetine is used as a non-stimulant. Atomoxetine was the first non-stimulant medication approved by the United States Food and Drug Administration (FDA) to treat ADHD patients aged over six years [9]. Methylphenidate is a short-acting stimulant that is well tolerated in the short term. It reduces height and weight in the long term [10,11], while the short-term effects of atomoxetine on growth development are not well documented. Long-term atomoxetine treatment may be associated with deficits in height growth [10]. In addition, some studies have indicated that atomoxetine might have additional side effects beyond norepinephrine reuptake inhibition. Hence, the safety and efficacy of this treatment require further investigation [11,12]. Therefore, this study was conducted to investigate the short-term adverse effects of atomoxetine, and its potential impacts on height and weight changes in primary school children.

Studies performed during the last decade indicate that the development of ADHD involves a complex interaction of the neurobiological and neurochemical systems. However, the precise etiology and pathophysiology behind this disease are still not fully understood [8,13,14]. Neurotrophins, which include brain-derived neurotrophic factor (BDNF), nerve growth factor (NGF), and insulin-like growth factor-1 (IGF-1), are a family of proteins that regulate neural growth, survival, and differentiation [15,16]. Recently, animal and clinical studies have indicated that the brain-derived neurotrophic factors involved the pathophysiology of ADHD and could be potential biomarkers [17,18]. IGF-1 is one of the more frequently investigated neurotrophins. It is the principal mediator of growth hormone (GH), and plays a critical role in regulating both anabolic and catabolic pathways in skeletal muscle, and in promoting bone formation and growth [19,20]. In addition, the IGF-1/IGFBP-3 axis was considered to be a potential biochemical growth maturity indicator, especially in children [21,22,23,24]. Previous studies applied IGF-1, IGFBP-3, and thyroid function tests as growth parameters to quantify the effects of methylphenidate on children’s growth development [25,26,27]. However, few studies have explored the impact of atomoxetine on children’s growth development according to these three growth parameters. Hence, this study investigated the changes in serum IGF-1, IGFBP-3 levels, thyroid function, total protein (TP), albumin (ALB), and hemoglobin (HGB) to quantify the effects of atomoxetine on children’s growth development in the short term.

Although it is well known that primary school children with ADHD often present with difficulties in the morning prior to school and in the evening, these two periods, and the family interactions that they entail, are often neglected in studies of ADHD [28,29]. A novel phase III delayed release and extended release methylphenidate (HLD200) was designed to provide an extended duration of release from the morning prior to school, throughout the day, and into the evening [28,30,31,32]. Kelsey et al.’s study indicated that a once-daily administration of atomoxetine in the morning provided safe, rapid, and continuous symptom relief throughout the day [33]. However, information about the effects of atomoxetine on different periods of the day is limited. Questionnaire–Children with Difficulties (QCD) has been widely used in China to evaluate parents’ perceptions of ADHD and patients’ daily behaviors during different times of the day, such as in the morning, during school, after school, in the evening, and at night time [34,35]. Therefore, this study first used the QCD to quantify the efficacy of atomoxetine treatment in improving behavioral difficulties at different times.

## 2. Materials and Methods

### 2.1. Data Collection

This prospective study was conducted at the Department of Pediatrics at the Affiliated Hospital of Jiangnan University from August 2019 to February 2021. The criteria for study inclusion were: (1) aged between 6 and 12 years; (2) newly diagnosed ADHD patients; (3) met the ADHD diagnostic criteria via clinical assessment according to the *Diagnostic and Statistical Manual of Mental Disorders, Fifth Edition* (DSM-V); (4) received a 12-week treatment with atomoxetine for the first time. Patients with developmentally induced growth suppression disorders, congenital malformations, or chronic somatic diseases, such as asthma, arthritis, and chronic back/neck pain, were excluded from the study. Patients suffering from neural disorders (including epilepsy, cerebral palsy, or developmental delay) and acute diseases at the time of admission (including upper respiratory tract infection and enteritis) were also excluded. Patients who received previous pharmacological treatment with either atomoxetine or methylphenidate, or others, were not included. The workflow used to screen for eligible participants is illustrated in Figure 1.

### 2.2. Ethical Considerations

The guardians or parents of the patients participating in the study provided written informed consent for us to obtain anonymous data from the children’s medical records at the very beginning. They were also fully informed that thyroid function, serum IGF-1, and IGFBP-3 levels were measured for the clinical assessment of growth development and for the purpose of clinical research. The Research Ethics Committees of the Affiliated Hospital of Jiangnan University (Wuxi, China) approved the study (No.LS2019047).

### 2.3. Treatment

The children received pharmacologic therapy as indicated by the second edition of the Chinese ADHD guidelines, released in 2015 [36]. Atomoxetine (Eli Lilly and Company, Indianapolis, IN, USA) was administered orally once daily (d) using an initial dose of 0.5 mg/kg/d. The amount was gradually increased up to a maximum of 1.2 mg/kg/d according to the tolerance of the drug and the improvement of core symptoms. Once symptom control was achieved, the dose was maintained throughout the treatment.

### 2.4. Questionnaires

To quantify the effects of atomoxetine at different periods of a day, and its efficacy at alleviating the core symptoms of ADHD, an inventory of the improvement in the children’s behavior and core symptoms of ADHD was kept by using three parent-reported questionnaires, completed electronically, including the Chinese version of the Questionnaire–Children with Difficulties (QCD), the Chinese version of the Swanson, Nolan, and Pelham IV scale (SNAP-IV), and the Conners’ Parents Rating Scales (CPRS). The QCD scale consists of 20 questions that focus on the difficulties faced by children with ADHD before going to school (Q 1~4), during school (Q 5~7), after school (Q 8~10), in the evening (Q 11~14), and at night (Q 15~18), and on their overall behavior (Q 19~20). Four different level scores were used: 0 means disagree, 1 means partially agree, 2 means mostly agree, and 3 means complete agree. This questionnaire is a reliable and valid tool with which to assess daily ADHD symptoms in various Chinese studies [35,37]. SNAP-IV is based on the DSM-5 criteria for ADHD, and the Chinese SNAP-IV can accurately assess three presentation specifiers (inattention, hyperactivity/impulsivity, and oppositional), as can its English version [38]. This tool is widely used to evaluate the effectiveness of ADHD treatments, and was found to be a reliable and well-validated instrument in previous studies [39,40]. CPRS is a parent-reported tool used to quantitatively measure specific behavioral, social, and academic issues in children aged between 6 and 18 years [41]. This tool is also commonly used to diagnose ADHD, and assesses the impact of ADHD symptoms on the children’s academic performance, behavior, quality of life, and relationships [42,43].

### 2.5. Adverse Reactions and Serum Analysis

To investigate the possible adverse reactions to atomoxetine, an adapted semi-structured interview was recorded every three weeks by the guardians or parents electronically (Appendix A) based on the literature [44,45,46]. In this study, we focused on the most common drug adverse reactions, such as severe adverse reactions (headache, insomnia, suicide attempt) and gastrointestinal symptoms (loss of appetite, constipation, vomiting), fatigue, dizziness, and emotional instability) [47,48]. To quantify the effects of atomoxetine on children’s growth development in the short term, thyroid function, TP, ALP, and HGB levels were measured pre-treatment and post-treatment. The serum levels of IGF-1 and IGFBP-3 was also measured via the chemiluminescent immunometric assay (CLIA) method and analyzed in our hospital’s laboratory [49].

### 2.6. Statistical Analyses

Statistical analyses were performed using the Statistical Package for the Social Sciences (SPSS) software version 23.0 (IBM Corporation, Armonk, NY, USA). The results were presented as means ± standard deviations. The pre-treatment and post-treatment variables were compared using a paired *t*-test. For all tests, a *p*-value below 0.05 was considered to be statistically significant.

## 3. Results

### 3.1. Demographics

A total of 155 patients chose atomoxetine treatment via a shared decision-making process, with input from the physician and the guardians or parents. The parents/guardians of six children refused to provide their medical records regarding atomoxetine treatment. Finally, 149 children were enrolled in this study and completed 12 weeks of atomoxetine treatment, including 56 girls and 93 boys. The demographics and clinical characteristics are summarized in Table 1. The pre-treatment developmental status of all children included in the study was within the normal range, with an average age of 8.00 ± 1.29 years, an average height SDS (0.19 ± 0.77), weight SDS (−0.01 ± 0.35), and BMI SDS (−0.11 ± 0.46).

### 3.2. Questionnaires Results

#### 3.2.1. QCD

The results of the fully completed patient-reported QCDs at baseline and during the six different treatment time points are summarized in Table 2. All the six different QCD periods and total scores were significantly lower when compared with the baseline (*p* < 0.001). The effect sizes of the atomoxetine treatment, assessed by measuring the difference in the QCD scores between baseline and during the six different treatment time points, are also provided (morning r = 0.66, during school r = 0.57, after school r = 0.42, evening r = 0.73, night r = 0.25, overall behavior r = 0.47, total score r = 0.83). Our results indicate that the atomoxetine treatment could significantly alleviate behavioral difficulties in primary children with ADHD, especially in the morning prior to school and in the evening.

#### 3.2.2. SNAP-IV

The SNAP-IV scores are summarized in Table 3. The SNAP-IV inattention mean score was 13.6 (SD = 5.8) at baseline and 6.1 (SD = 4.2) at 12 weeks. The SNAP-IV hyperactivity mean score was 11.7 (SD = 5.27) at baseline and 3.9 (SD = 1.59) at 12 weeks. SNAP-IV oppositional mean score was 7.9 (SD = 4.56) at baseline and 1.1 (SD = 0.35) at 12 weeks. The SNAP-IV total mean score was 33.2 (SD = 8.68) at baseline and 11.1 (SD = 4.38) at 12 weeks. The SNAP-IV sub-scores and global scores decreased significantly during the 12 weeks of treatment when compared with the baseline (*p* < 0.05). The effect sizes of the atomoxetine treatment measured the difference in the SNAP-IV scores between baseline and after treatment (inattention r = 0.59, hyperactivity/impulsivity r = 0.7, oppositional r = 0.72, total score r = 0.84).

#### 3.2.3. CPRS

The CPRS scores are summarized in Table 4. The conduct problems mean score was 0.83 (SD = 0.31) at baseline and 0.79 (SD = 0.29) at 12 weeks. The CPRS learning problem mean score was 1.52 (SD = 0.32) at baseline and 0.59 (SD = 0.35) at 12 weeks. The CPRS impulse-hyperactivity mean score was 1.19 (SD = 0.46) at baseline and 0.86 (SD = 0.4) at 12 weeks, and the hyperactivity index mean score was 1.10 (SD = 0.41) at baseline and 0.21 (SD = 0.18) at 12 weeks. The sub-scores for conduct problems, learning problems, impulse-hyperactivity, and hyperactivity index decreased significantly after treatment (*p* < 0.05), while the effect sizes of the atomoxetine treatment were r = 0.06, r = 0.81, r = 0.35, and r = 0.81, respectively.

### 3.3. Effects of Atomoxetine on Children’s Development and Serum Growth Parameters

The serum IGF-1 and IGFBP-3 levels were normal for their sex and age (Appendix A). The height SDS, weight SDS, BMI SDS, and serum growth parameters are illustrated in Table 5. There was a statistically significant difference in weight SDS after treatment when compared with the baseline (*p* < 0.05), but the effect size is slight (r = 0.09). However, the height SDS, BMI SDS, and the levels of serum growth parameters did not differ significantly from the baseline after 12 weeks of treatment (*p* > 0.05).

### 3.4. Adverse Reactions Results

Over the course of the 12 weeks, no severe adverse reactions, such as suicide attempt or suicidal ideation, were reported. Gastrointestinal adverse reactions were reported by 56.38% of participants during the first three weeks of treatment. The most common gastrointestinal symptom was the loss of appetite or anorexia (35.57%), followed by constipation (12.08%) and nausea and vomiting (8.72%). The incidence of neurological side effects was 24.16%, with lethargy (13.42%) being the most common symptom reported, followed by dizziness (6.71%) and emotional instability (4.03%). These adverse reactions were effectively alleviated by changing the work and rest time, the medicine administration time, the prandial time, and the diet. Furthermore, the incidence of adverse reactions decreased significantly in the later stage of the treatment, as shown in Table 6.

## 4. Discussion

The current prospective study investigated the improvement of behavioral difficulties at different times after a 12-week atomoxetine treatment for primary children diagnosed with ADHD aged between 6 and 12 years, and the possible influence on the growth parameters of this treatment. Although behavioral and psychosocial treatments were recommended before pharmacological therapy by the United Kingdom and Germany [50,51], effective behavioral and psychosocial treatments are still more expensive, time-consuming, and less available than pharmacological treatment in medically underserved areas [5]. There are only two drugs, methylphenidate and atomoxetine, approved as the first line to treat ADHD by the China Food and Drug Administration (CFDA, https://www.nmpa.gov.cn, accessed on 16 October 2021). At the same time, clonidine and bupropion were recommended as the second-line medications [36]. As a non-stimulant, atomoxetine works by inhibiting the reuptake of norepinephrine, and effectively increases norepinephrine concentration in the synaptic cleft [1]. To decide upon a first-line pharmacological treatment when choosing between methylphenidate and atomoxetine, the guardians or parents must be fully informed about the adverse reactions at first. This shared decision-making process needs physicians to cooperate with the guardians or parents. In China, the guardians or parents prefer to choose the atomoxetine treatment for the following reasons: (1) fewer adverse reactions on height; (2) they hope the effect of the drug will last throughout the day; (3) the ADHD symptoms are not severe and immediate onset is not needed; (4) concerns about drug abuse [5,52,53]. Short stature and growth deficits remain the most common pediatric concerns, and this is mainly presented with growth hormone (GH) deficiency accompanied by lower serum IGF-1 levels [54,55]. However, non-GH deficient short stature disorders, such as idiopathic short stature (ISS) and growth hormone insensitivity (GHI), often present with normal GH when accompanied by lower serum IGF-1 levels [56,57]. Hence, the Growth Hormone Research Society (GRS) and the Chinese official guidelines for diagnosing and treating children with short stature recommended IGF-1 and its main binding protein, IGFBP-3, as valuable biomarkers in assessing growth deficits [58,59]. Our prospective study indicated that the use of atomoxetine significantly reduced the symptoms and behavioral difficulties experienced by children who have ADHD. Furthermore, the treatment was generally well tolerated, with few adverse events, and was not associated with growth deficits in the short term.

To the best of our knowledge, this is the first report using the QCD to examine the efficiency of atomoxetine for children aged 6–12 years old with ADHD, in terms of reducing the functional difficulties at different times. The QCD was created by Yamashita, and is widely used in Asian countries, such as China and Japan [60]. The QCD has three key characteristics: the evaluation of life function, the assessment of life function at different periods of the day, and the convenience of use in daily clinical practice [34,35]. Compared with the Child Behavior Checklist (CBCL), which is often used in Western countries, QCD has shown its advantages because it includes only 20 practical and easy-to-understand questions, and only a short time is needed to complete the QCD [61]. The QCD also provides physicians with the necessary information for selecting appropriate drug therapy [34]. In this study, we focused on the two periods that have often been neglected in previous studies of ADHD. The effect of atomoxetine in the morning prior to school (*p* < 0.001, r = 0.66) and in the evening (*p* < 0.001, r = 0.73) was assessed by comparing QCD scores at different times. Our QCD results indicated that atomoxetine could significantly reduce the functional difficulties experienced by ADHD children in these two periods. Family interactions were another neglected point in previous studies of ADHD [29]. Although it is well known that primary school children with ADHD often present with difficulties in the morning prior to school and in the evening [28], previous studies have indicated that parents find it more challenging to take care of ADHD children in the evening. They have reported difficulties in persuading their children to complete their homework and enjoy family time [62,63]. Hence, alleviating the core symptoms of ADHD in the morning prior to school and in the evening may effectively solve this problem and reduce parental stress levels. Minimizing parenting stress and improving family interactions are important because they can reduce the risk of adverse outcomes in the long term [64,65,66]. Consistent with our QCD results, atomoxetine treatment may be a choice in China to alleviate parental stress and to improve family interactions in children with ADHD. This study also compared the SNAP-IV scale and CPRS after 12 weeks of treatment with atomoxetine, which further confirmed that atomoxetine is an effective treatment for primary school children with ADHD. The SNAP-IV results showed a significant improvement in the core symptoms of ADHD, while the CPRS results indicated a significant improvement in controlling ADHD symptoms across two different domains, learning problems (r = 0.81) and hyperactivity (r = 0.86). Consistent with previous studies [67,68,69], our findings indicate that atomoxetine can reduce the core symptoms in primary school children with ADHD.

In this study, the levels of serum growth parameters were closely monitored throughout treatment. These markers have essential roles in promoting children’s growth and development [70,71,72]. In contrast to the BDNF that was lower in newly diagnosed ADHD patients [73], this study found that the serum levels of IGF-1 and IGFBP-3 were within the normal range in newly diagnosed primary school patients with ADHD. These results indirectly confirmed that the secretion and role of endogenous growth hormones were normal in newly diagnosed primary school patients with ADHD. In addition, there was no difference in the levels of IGF-, IGFBP-3, and thyroid function before and after 12 weeks of atomoxetine treatment. A mice model study also showed that atomoxetine did not reduce the expression of the IGF-1 gene [74]. Our serum biochemical results suggested that the atomoxetine did not affect the level of serum growth parameters in children. However, whether the treatment of atomoxetine will affect the levels of serum growth parameters in primary school children with ADHD in the long term remains to be further studied and discussed.

Our study reported no severe adverse reactions, such as suicide attempt or suicidal ideation, throughout treatment. The most significant initial adverse reactions were gastrointestinal symptoms, including the loss of appetite or anorexia, constipation, nausea, and vomiting. These symptoms might be caused by temporary disorders of the central norepinephrine system associated with hunger or satiety [28]. Interestingly, there was no significant difference in the levels of ALB, HGB, and HCT after treatment compared with the baseline. This indicated that, although atomoxetine could cause gastrointestinal adverse reactions, it did not affect the nutritional status of ADHD children. In addition, the height and BMI of children did not lag behind children of the same age without ADHD, and the serum growth parameters were also normal. In our study, despite a significant weight reduction being noted after treatment (*p* < 0.05), the effect size was slight (r = 0.09). A survey by Germinario revealed that the height and weight gradually returned to normal after two years of ADHD treatment. This improvement was more evident for atomoxetine when compared with treatment using methylphenidate [75]. Consistent with our previous results, we speculated that atomoxetine might lead to temporary weight loss due to a loss of appetite. This factor may affect the selection of the short-term pharmacological treatment in China. Based on the guidelines from National Institute for Health and Care Excellence (NICE), temporary weight loss does not require particular caution. If weight loss is of clinical concern, medication should be taken either during or after meals rather than before meals, and additional meals or snacks early in the morning or late in the evening are also suggested [51].

## 5. Limitations

Our study has some limitations that have to be acknowledged. First of all, all enrolled patients were from a single center, so a selection bias could not be entirely ruled out. Secondly, although all questionnaires were completed by parents/guardians electronically at the clinical department and there were no missing data, the expectations of parents/guardians and their insufficient training by the clinicians might have introduced a positive bias into our findings. Furthermore, although all enrolled patients were newly diagnosed with ADHD, this study did not compare the use of atomoxetine with a placebo. Therefore, the symptom improvement noted in ADHD children could have resulted from natural growth and development. Finally, the small number of patients enrolled in this study may affect the generalizability of the research findings.

## 6. Conclusions

In summary, this study determined that atomoxetine positively affects behavioral difficulties in primary school children with ADHD and confirmed that atomoxetine would not affect growth development in the short term. In addition, our study was the first one using the QCD to prove that atomoxetine treatment can significantly reduce the functional difficulties experienced by ADHD children in the morning prior to school and in the evening. Atomoxetine may be a good choice in China to alleviate the stress of parents/guardians, and to improve family interactions with primary school children with ADHD. Moreover, our study indicated that the serum levels of IGF-1 and IGFBP-3 were within the normal range in newly diagnosed ADHD patients, and that atomoxetine will not affect the serum concentration of growth parameters in the short-term.

## Figures and Tables

**Figure 1 children-09-00212-f001:**
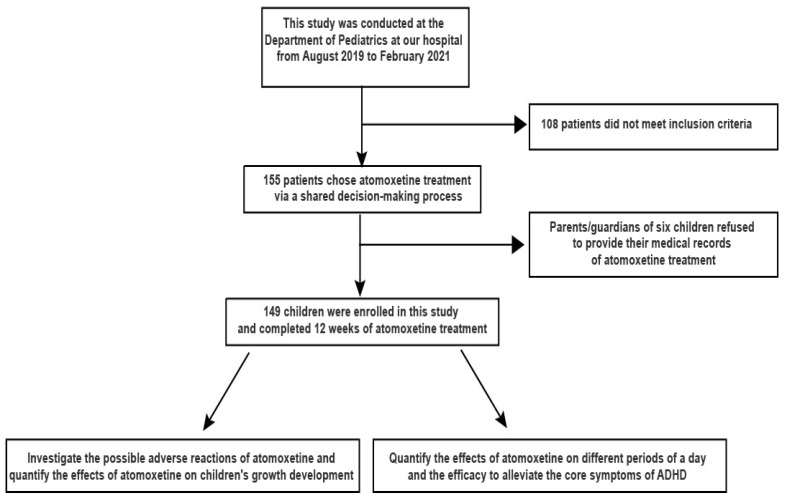
Flow chart of the study.

**Table 1 children-09-00212-t001:** Demographic and clinical characteristics of patients.

Characteristics	Mean (SD)/N (%)
Age (years, mean (SD))	8.9 (2.78)
Gender, boys, N (%)	93 (62%)
Gender, girls, N (%)	56 (38%)
ADHD subtype, N (%)	
Predominantly inattentive presentation	81 (54%)
Predominantly hyperactive-impulsive presentation	5 (3%)
Combined presentation	63 (43%)
Any psychiatric comorbidities	
Tic disorder	9 (6%)
ODD/CD	5 (3%)
Anxiety	4 (2%)

Abbreviations: ODD: obsessive compulsive disorder; CD: conduct disorder.

**Table 2 children-09-00212-t002:** QCD score.

QCD	Baseline		12 Weeks		t	*p* Value	Cohen’s *d*	Effect Size r
	Mean	SD	Mean	SD				
Morning **	5.12	1.19	7.10	1.02	−17.45	<0.001	1.78	0.66
During school **	5.58	1.07	7.03	0.97	−14.16	<0.001	1.41	0.57
After school **	6.18	1.05	7.18	1.10	−8.91	<0.001	0.92	0.42
Evening **	5.75	1.00	7.88	0.98	−17.23	<0.001	2.15	0.73
Night **	7.53	1.08	8.08	0.96	−5.86	<0.001	0.53	0.25
Overall behavior **	3.00	1.04	4.05	0.89	−9.06	<0.001	1.08	0.47
Total score **	33.15	2.64	41.30	2.73	−22.75	<0.001	3.03	0.83

** *p* < 0.001. Abbreviations: QCD: Questionnaire–Children with Difficulties.

**Table 3 children-09-00212-t003:** SNAP-IV score.

Project	Baseline		12 Weeks		t	*p* Value	Cohen’s *d*	Effect Size r
	Mean	SD	Mean	SD				
Inattention **	13.60	5.80	6.10	4.20	6.18	<0.001	1.48	0.59
Hyperactivity/impulsivity **	11.70	5.27	3.90	1.59	4.46	<0.001	2.0	0.7
Oppositional **	7.90	4.56	1.10	0.35	5.95	<0.001	2.1	0.72
Total score **	33.20	8.68	11.10	4.38	8.50	<0.001	3.2	0.84

** *p* < 0.001. Abbreviations: SNAP-IV: The Swanson, Nolan, and Pelham scale version IV.

**Table 4 children-09-00212-t004:** CPRS score.

Project	Baseline		12 Weeks		t	*p*-Value	Cohen’s *d*	Effect Size r
	Mean	SD	Mean	SD				
Conduct problems *	0.83	0.31	0.79	0.29	2.49	0.017	0.13	0.06
Learning problems **	1.52	0.32	0.59	0.35	14.30	<0.001	2.77	0.81
Psychosomatic disorders	0.20	0.27	0.19	0.24	1.43	0.16	0.03	0.019
Impulse-hyperactivity **	1.19	0.46	0.86	0.40	6.84	<0.001	0.76	0.35
Anxiety	0.38	0.30	0.31	0.21	3.09	0.06	0.27	0.13
Hyperactivity index **	1.10	0.41	0.21	0.18	5.80	<0.001	2.8	0.81

* *p* < 0.05; ** *p* < 0.001. Abbreviations: CPRS: Conners’ Parents Rating Scales.

**Table 5 children-09-00212-t005:** Comparison of height, weight, BMI, and serum growth parameters.

	Baseline		12 Weeks		t	*p*	Cohen’s *d*	Effect Size r
	Mean	SD	Mean	SD				
Height SDS	0.19	0.77	0.08	0.80	2.80	0.08	0.14	0.06
Weight SDS *	−0.01	0.35	−0.08	0.37	−2.58	0.03	0.19	0.09
BMI SDS	−0.11	0.46	−0.12	0.50	0.24	0.82	0.02	0.01
IGF-1, ng/mL	170.43	37.27	209.71	83.53	−1.20	0.28	0.60	0.27
IGFBP-3, μg/mL	4.22	0.87	4.62	0.98	−7.00	0.51	0.43	0.21
T3, ng/dL	115.1	10.56	114.71	14.45	0.07	0.95	0.03	0.01
T4, ug/dL	8.35	1.54	8.07	0.60	0.53	0.61	0.23	0.12
TSH, mIU/mL	2.64	1.46	2.44	1.00	0.54	0.60	0.15	0.07
TP, g/L	71.76	3.73	72.42	3.55	−0.44	0.67	0.18	0.09
ALB, g/L	45.53	1.68	45.61	1.86	−0.12	0.90	0.04	0.02
HGB, g/L	133.91	5.87	137.09	6.99	−1.31	0.22	0.49	0.23

* *p* < 0.05. Abbreviations: BMI: body mass index; SDs: standard deviation scores; IGF-1: insulin-like growth factor-1; IGFBP-3: insulin-like growth factor-binding protein 3; T3: triiodothyronine, T4: thyroxine; TSH: thyroid-stimulating hormone; TP: total protein; ALB: albumin; HGB: hemoglobin.

**Table 6 children-09-00212-t006:** Incidence of adverse reactions, N (%).

Time (Weeks)	Loss of Appetite or Anorexia	Lethargy	Constipation	Sickness and Vomiting	Dizziness	Emotional Instability
3	53 (36%)	20 (13%)	18 (12%)	13 (8%)	10 (6%)	6 (4%)
6	40 (27%)	15 (10%)	8 (5%)	0	5 (3%)	0
9	28 (19%)	5 (3%)	0	0	0	0
12	10 (7%)	5 (3%)	0	0	0	0

## Data Availability

The data used to support the conclusions of this article are available from the corresponding authors upon request.

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
