# Peer review of "Effect of Atomoxetine on Behavioral Difficulties and Growth Development of Primary School Children with Attention-Deficit/Hyperactivity Disorder: A Prospective Study"

_children, 2022, doi:10.3390/children9020212_

Round 1
Reviewer 1 Report
Thanks for giving me the opportunity of reviewing this manuscript.
This is a clinical study of retrospective chart review to examine the effectiveness and safety of atomoxetine for treatment of ADHD in children ages of 6 and 12 years.
It is critical to obtain further information on the effectiveness and safety of any treatment agent for treatment of psychopathologies in pediatric population, since data are lacking in any area significantly.
That being said, there are major concerns on the writing, design, and results of the study that require at least significant revision (see blow for the details) and I cannot recommend publication of the manuscript as the current form.
Major Points
Introduction
Overall, the argument for the need of this type of study is very week. Why it would be important to look into IGF-1? What they also looked into IGFBP3? What is the reason for looking at the diurnal variation of the symptoms and its relation to atomoxetine treatment? What is the current understanding of the diurnal variation of ADHD symptoms? What’s the current understanding of impacts of other treatment modalities (for example, stimulants) on the diurnal variation of ADHD symptoms? What is the current understanding of the side effect profiles of atomoxetine (including weight and height changes)? The authors need to take further effort to summarize these findings, to make stronger argument for the need of this type of study. Significant revision is warranted.
Materials and Methods
When did the consenting procedure occur? Since it is a retrospective chart-review study, I was not sure when they obtained the consents from the parents. Also, was there any reason not to obtain assents from the patients themselves? And I was not sure the Research Ethics Committee is equivalent to the Institutional Review Board?
How was the clinical decision made to use atomoxetine? Usually stimulants (methylphenidate or dextroamphetamine) are prescribed as first line agent, so I was not sure why these children received atomoxetine without previous history of treatment by, say, methylphenidate.
Page 3 Line 129: Was blood test of IGF-1 and IGFBP-3 a part of regular clinical treatment? I was very confused since this is a retrospective chart review study, and these are rarely a part of standard care, so how they were obtained. The authors need to provide further explanation on this.
Results
Page 3 Line 139: What does this mean to have 263 children included and 149 enrolled? It is very confusing what this means exactly.
Overall the lack of control group is a significant issue of the study to interpret the results. How to control the placebo effect?
Page 4 Line 155: How did the authors decide one effect size is significantly bigger than others? Or did the authors compare the degree of effect size at all? Otherwise they can argue atomoxetine relieves symptoms more so in the evenings.
Discussion
Why the authors think atomoxetine works better for symptom relief in the evenings? And to have this conclusion, what type of statistical analysis was done (comparison of effect size in different times)? What is the clinical implication of this finding?
Minor points
Abstract
Page 1 Line 17: “china” should be spelled as “China”.
Page 1 Line 18: “Despite previous studies…”: This sentence does not make any sense in its grammar. Should be something like “The efficacy and safety of atomoxetine has been established…”.
Page 1 Result: It did not specify what specific time of the day the symptoms improved and which symptoms improved, which were the main goals of the study.
Page 1 Line 38: “It can effectively can reduce…”. Again, grammatically wrong. This is very recurring problem in the manuscript, and I highly encourage the authors thoroughly re-examine and revise the manuscript for its grammar errors.
Introduction
Page 2 Line 58: ADHD is not a neurodegenerative disorder so this sentence is irrelevant to the study.
Page 2 Line 64: “central stimulants” is not a common term used. I assumed they authors meant to say “central nervous system stimulants”?
Page 2 Line 72: “Despite previous studies…” Again, this sentence does not make sense in its grammar.
Materials and Methods
Page 2 Line 84: “via clinically assessed” is not right in its grammar. It should be “via clinical assessment”.
Page 2 Line 87: What were the examples of the “chronic somatic diseases”?
Page 2 Line 90: What about other forms of treatment for ADHD? Such as dextroamphetamine, lisdextroamfetamine, guanfacine, clonidine, or TCAs? Were they included?
Discussion
Page 8 Line 253-254: You cannot argue the findings of an animal model is “the same” as a human study.
Page 8 Line 272: Is there any clinical implication of the “temporary weight loss”? Any guideline on how to address this?
Author Response
Major Points
Introduction
Q1: Overall, the argument for the need of this type of study is very week. Why it would be important to look into IGF-1? What they also looked into IGFBP3?
Response: We are very sorry for this confusion. Recent animal and clinical studies indicated that neurotrophins such as brain-derived neurotrophic factors involved the pathophysiology of ADHD and could be potential biomarkers [1, 2]. IGF-1 is one of the frequently investigated neurotrophins. It is the principal mediator of growth hormone (GH) and plays a critical role in regulating both anabolic and catabolic pathways in skeletal muscle, and promoting bone formation and growth. In addition, the IGF-1/ IGFBP-3 axis was considered a potential biochemical growth maturity indicator, especially in children [3-6]. Previous studies applied IGF-1, IGFBP-3, and thyroid function tests as growth parameters to quantify the effects of methylphenidate on children's growth development [7-9]. However, few studies explored the impact of atomoxetine on children's growth development by these three growth parameters. Hence, this retrospective study investigated the changes in serum IGF-1, IGFBP-3 levels, thyroid function tests. We added this information in the introduction part. Please see the highlighted parts in the revised manuscript.
Q2: What is the reason for looking at the diurnal variation of the symptoms and its relation to atomoxetine treatment? What is the current understanding of the diurnal variation of ADHD symptoms? What’s the current understanding of impacts of other treatment modalities (for example, stimulants) on the diurnal variation of ADHD symptoms?
Response: We are very sorry for this confusion. Although it is well known that primary school children with ADHD often present with difficulties in the morning prior to school and in the evening [10]. These two periods and family interactions were often neglected in previous studies of ADHD [11]. A novel phase III delayed-release and extended-release methylphenidate (HLD200) was designed to provide an extended duration from the morning prior to school, throughout the day, and into the evening [10, 12-14]. Kelsey et al.’s study indicated that once-daily administration of atomoxetine in the morning provided safe, rapid, and continuous symptom relief throughout the day [15]. However, information about the effects of atomoxetine on different periods of the day is limited. Therefore, this study first used QCD to retrospectively quantify the efficacy of atomoxetine treatment in improving behavioral difficulties at different times. We added this information in the introduction part. Please see the highlighted parts in the revised manuscript.
Q3: What is the current understanding of the side effect profiles of atomoxetine (including weight and height changes)?
Response: Based on the reviewer’s suggestion, we add this information: Methylphenidate is a short-acting stimulant with well-tolerance in the short-term. It reduces the height and weight in the long-term [16], while the short-term effects of atomoxetine on growth development are not well documented. Long-term atomoxetine treatment may be associated with deficits in height growth [17, 18]. Please see the highlighted parts in the revised manuscript.
Materials and Methods
Q4: When did the consenting procedure occur? Since it is a retrospective chart-review study, I was not sure when they obtained the consents from the parents. Also, was there any reason not to obtain assents from the patients themselves? And I was not sure the Research Ethics Committee is equivalent to the Institutional Review Board?
Response: We are very sorry for this confusion. Actually, the consenting procedure was conducted at the very beginning, prior to the start of this chart-review study. The reason why our final approval date seems conflicting is due to the relocation of our hospital. Our hospital official website could confirm this relocation(http://www.wuxihospital.com/info/1034/2093.htm, in Chinese). As the patients are minors, under age 18 years old, therefore, consents from their parent or other guardians are required according to local law and regulations for any research and treatment for non-emergent/non-life-threatening illness. Most Chinese retrospective study involving children prefer to obtain the consents from the parents [19-22]. The Research Ethics Committee is equivalent to the Institutional Review Board, and this could be confirmed in our previous studies [23-26].
Q5: How was the clinical decision made to use atomoxetine? Usually stimulants (methylphenidate or dextroamphetamine) are prescribed as first line agent, so I was not sure why these children received atomoxetine without previous history of treatment by, say, methylphenidate.
Response: Based on the reviewer’s suggestion, we add this information: To decide first-line pharmacological treatment between methylphenidate and atomoxetine, the guardians or parents must be fully informed about the adverse reactions first. This shared decision-making process needs physicians to cooperate with the guardians or parents. In China, the guardians or parents prefer to choose atomoxetine treatment for the following reasons: 1) adverse reactions on height; 2) they hope the effect of the drug last throughout the day; 3) the ADHD symptoms is not severe and immediate onset is not needed; 4) concerns about drug abuse [16, 27, 28]. Please see the highlighted parts in the revised manuscript.
Q6: Page 3 Line 129: Was blood test of IGF-1 and IGFBP-3 a part of regular clinical treatment? I was very confused since this is a retrospective chart review study, and these are rarely a part of standard care, so how they were obtained. The authors need to provide further explanation on this.
Response: We are very sorry for this confusion. Because adverse reactions on height is an important factor in China to select first-line agent and the IGF-1/ IGFBP-3 axis was considered a potential biochemical growth maturity indicator, especially in children [3-6]. The guardians or parents of patients are fully informed about the IGF-1/ IGFBP-3 test, as the result of share-decision making, the serum levels of IGF-1 and IGFBP-3 were also measured via the chemiluminescent immunometric assay (CLIA) method and analyzed in our hospital’s laboratory [29]. Please see the highlighted parts in the revised manuscript
Results
Q7: Page 3 Line 139: What does this mean to have 263 children included and 149 enrolled? It is very confusing what this means exactly. Overall the lack of control group is a significant issue of the study to interpret the results. How to control the placebo effect?
Response: We are very sorry for this confusion. We had redrawn the Figure 1 and this study included 263 newly diagnosed ADHD children, of which 155 patients chose atomoxetine treatment via a shared decision-making process with input from the physician and the guardians or parents. Parents/guardians of six children refused to provide their medical records of atomoxetine treatment. Finally, 149 children were enrolled in this study and completed 12 weeks of atomoxetine treatment. Based on the reviewer’s suggestion, we add this information in the discussion part: although all enrolled patients were newly diagnosed with ADHD, this study did not compare the use of atomoxetine with a placebo. Therefore, as a limitation of our study, the symptoms improvement noted in ADHD children could have resulted from natural growth and development. Please see the highlighted parts in the revised manuscript
Q8: Page 4 Line 155: How did the authors decide one effect size is significantly bigger than others? Or did the authors compare the degree of effect size at all? Otherwise they can argue atomoxetine relieves symptoms more so in the evenings.
Response: We are very sorry for this confusion. To make our result more convincing, we made corrections in this revised manuscript: In this retrospective study, we focus on the two periods that were often neglected in studies of ADHD, the effect size of atomoxetine on the morning prior to school (P < 0.001, r =0.66) and evening (P < 0.001, r =0.73) was calculated via comparing QCD scores at different times. Our QCD results in this study indicated that atomoxetine could significantly reduce the functional difficulties experienced by ADHD children in these two periods. Please see the highlighted parts in the revised manuscript
Discussion
Q9: Why the authors think atomoxetine works better for symptom relief in the evenings? And to have this conclusion, what type of statistical analysis was done (comparison of effect size in different times)? What is the clinical implication of this finding?
Response: We are very sorry for this confusion. To make our result more convincing, we made corrections in this revised manuscript: In this retrospective study, we focus on the two periods that were often neglected in studies of ADHD, the effect size of atomoxetine on the morning prior to school (P < 0.001, r =0.66) and evening (P < 0.001, r =0.73) was calculated via comparing QCD scores at different times. Our QCD results in this study indicated that atomoxetine could significantly reduce the functional difficulties experienced by ADHD children in these two periods. Although it is well known that primary school children with ADHD often present with difficulties in the morning prior to school and in the evening. Previous studies indicated that parents found it more challenging to take care of ADHD children in the evening. They reported difficulties persuading their children to do their homework and enjoy family time [33] Hence, alleviating the core symptoms of ADHD in the morning prior to school and in the evening may effective solve this problem and atomoxetine may be a good choice in China to alleviate the stress of parents/guardians and improve family interactions with primary school children with ADHD. Please see the highlighted parts in the revised manuscript.
Minor points
Abstract
Q10: Page 1 Line 17: “china” should be spelled as “China”.Page 1 Line 18: “Despite previous studies…”: This sentence does not make any sense in its grammar. Should be something like “The efficacy and safety of atomoxetine has been established…”.Page 1 Result: It did not specify what specific time of the day the symptoms improved and which symptoms improved, which were the main goals of the study.Page 1 Line 38: “It can effectively can reduce…”. Again, grammatically wrong. This is very recurring problem in the manuscript, and I highly encourage the authors thoroughly re-examine and revise the manuscript for its grammar errors.
Response: Based on the reviewer’s suggestion, we made the corrections in the abstract part. Please see the highlighted parts in the revised manuscript.
Introduction
Q11: Page 2 Line 58: ADHD is not a neurodegenerative disorder so this sentence is irrelevant to the study. Page 2 Line 64: “central stimulants” is not a common term used. I assumed they authors meant to say “central nervous system stimulants”? Page 2 Line 72: “Despite previous studies…” Again, this sentence does not make sense in its grammar.
Response: Based on the reviewer’s suggestion, we made the corrections in the introduction
part. Please see the highlighted parts in the revised manuscript
Materials and Methods
Q11: Page 2 Line 84: “via clinically assessed” is not right in its grammar. It should be “via clinical assessment”.Page 2 Line 87: What were the examples of the “chronic somatic diseases”?
Response: Based on the reviewer’s suggestion, we made the corrections in the revised manuscript. Chronic somatic diseases such as asthma, arthritis, and chronic back/neck pain were linked with mental disorders [33]and these patients were excluded from the study. Please see the highlighted parts in the revised manuscript
Q12: Page 2 Line 90: What about other forms of treatment for ADHD? Such as dextroamphetamine, lisdextroamfetamine, guanfacine, clonidine, or TCAs? Were they included?
Response: We are very sorry for this confusion, there are only two drugs-methylphenidate and atomoxetine approved as the first-line to treat ADHD by China Food and Drug Administration (CFDA, https://www.nmpa.gov.cn). At the same time, clonidine and bupropion were recommended as the second-line [34]. Patients who received other forms of treatment for ADHD were excluded from the study. Please see the highlighted parts in the revised manuscript
Discussion
Q13: Page 8 Line 253-254: You cannot argue the findings of an animal model is “the same” as a human study. Page 8 Line 272: Is there any clinical implication of the “temporary weight loss”? Any guideline on how to address this?
Response: Based on the reviewer’s suggestion, we made the corrections in the revised manuscript. Temporary weight loss may affect the selection of pharmacological treatment in the short-term in China. Based on the guidelines from National Institute for Health and Care Excellence (NICE), temporary weight loss does not require particular caution, and if weight loss is of clinical concern, medication should be taken either during or after meals rather than before meals, and additional meals or snacks early in the morning or late in the evening is also suggested [35]. Please see the highlighted parts in the revised manuscript
- Meng, W.D., et al., Elevated Serum Brain-Derived Neurotrophic Factor (BDNF) but not BDNF Gene Val66Met Polymorphism Is Associated with Autism Spectrum Disorders. Mol Neurobiol, 2017. 54(2): p. 1167-1172.
- Chang, J.P., et al., Cortisol, inflammatory biomarkers and neurotrophins in children and adolescents with attention deficit hyperactivity disorder (ADHD) in Taiwan. Brain Behav Immun, 2020. 88: p. 105-113.
- Bakker, N.E., et al., IGF-1 Levels, Complex Formation, and IGF Bioactivity in Growth Hormone-Treated Children With Prader-Willi Syndrome. J Clin Endocrinol Metab, 2015. 100(8): p. 3041-9.
- Jain, N., et al., Serum IGF-1, IGFBP-3 and their ratio: Potential biochemical growth maturity indicators. Prog Orthod, 2017. 18(1): p. 11.
- Sinha, M., et al., Serum and urine insulin-like growth factor-1 as biochemical growth maturity indicators. Am J Orthod Dentofacial Orthop, 2016. 150(6): p. 1020-1027.
- Sun, M., et al., Association Between Serum Calcium and Phosphorus Levels and Insulin-Like Growth Factor-1 in Chinese Children and Adolescents with Short Stature. Int J Gen Med, 2020. 13: p. 1167-1173.
- Bereket, A., et al., Height, weight, IGF-I, IGFBP-3 and thyroid functions in prepubertal children with attention deficit hyperactivity disorder: effect of methylphenidate treatment. Horm Res, 2005. 63(4): p. 159-64.
- Kim, W.J., et al., Preliminary Investigation of Association between Methylphenidate and Serum Growth Markers in Children with Attention-Deficit/Hyperactivity Disorder: A Cross-Sectional Case-Control Study. Soa Chongsonyon Chongsin Uihak, 2020. 31(3): p. 154-160.
- Toren, P., et al., Lack of effect of methylphenidate on serum growth hormone (GH), GH-binding protein, and insulin-like growth factor I. Clin Neuropharmacol, 1997. 20(3): p. 264-9.
- Wilens, T.E., et al., Clinically Meaningful Improvements in Early Morning and Late Afternoon/Evening Functional Impairment in Children with ADHD Treated with Delayed-Release and Extended-Release Methylphenidate. J Atten Disord, 2022. 26(5): p. 696-705.
- Faraone, S.V., et al., Functional Impairment in Youth With ADHD: Normative Data and Norm-Referenced Cutoff Points for the Before School Functioning Questionnaire and the Parent Rating of Evening and Morning Behavior Scale, Revised. J Clin Psychiatry, 2019. 81(1).
- Liu, T., et al., Pharmacokinetics of HLD200, a Delayed-Release and Extended-Release Methylphenidate: Evaluation of Dose Proportionality, Food Effect, Multiple-Dose Modeling, and Comparative Bioavailability with Immediate-Release Methylphenidate in Healthy Adults. J Child Adolesc Psychopharmacol, 2019. 29(3): p. 181-191.
- Pliszka, S.R., et al., Efficacy and Safety of HLD200, Delayed-Release and Extended-Release Methylphenidate, in Children with Attention-Deficit/Hyperactivity Disorder. J Child Adolesc Psychopharmacol, 2017. 27(6): p. 474-482.
- Childress, A.C., et al., A Randomized, Double-Blind, Placebo-Controlled Study of HLD200, a Delayed-Release and Extended-Release Methylphenidate, in Children with Attention-Deficit/Hyperactivity Disorder: An Evaluation of Safety and Efficacy Throughout the Day and Across Settings. J Child Adolesc Psychopharmacol, 2020. 30(1): p. 2-14.
- Kelsey, D.K., et al., Once-daily atomoxetine treatment for children with attention-deficit/hyperactivity disorder, including an assessment of evening and morning behavior: a double-blind, placebo-controlled trial. Pediatrics, 2004. 114(1): p. e1-8.
- Carucci, S., et al., Long term methylphenidate exposure and growth in children and adolescents with ADHD. A systematic review and meta-analysis. Neurosci Biobehav Rev, 2021. 120: p. 509-525.
- Kweon, K., et al., Effects of Atomoxetine on Height and Weight in Korean Children and Adolescents with Attention-Deficit/Hyperactivity Disorder: A Retrospective Chart Review. Psychiatry Investig, 2018. 15(6): p. 649-654.
- Corona, J.C., et al., Atomoxetine produces oxidative stress and alters mitochondrial function in human neuron-like cells. Sci Rep, 2019. 9(1): p. 13011.
- Wang, R., et al., Clinical and molecular features of children with Beckwith-Wiedemann syndrome in China: a single-center retrospective cohort study. Ital J Pediatr, 2020. 46(1): p. 55.
- Xu, Y.Y., et al., Retrospective analysis of epidemic thunderstorm asthma in children in Yulin, northwest China. Pediatr Res, 2021. 89(4): p. 958-961.
- Fang, Z., et al., Characteristics and outcomes of children with dissociative (conversion) disorders in western China: a retrospective study. BMC Psychiatry, 2021. 21(1): p. 31.
- Qiu, H., et al., Clinical and epidemiological features of 36 children with coronavirus disease 2019 (COVID-19) in Zhejiang, China: an observational cohort study. Lancet Infect Dis, 2020. 20(6): p. 689-696.
- Tianxiao Li, R.X., Chunhong Wang, Chenyu sun, Yueying Liu. Effectiveness of Recombinant Human Growth Hormone Therapy for Children with Phelan-McDermid Syndrome: A Randomized, Cross-over, Placebo-controlled, Preliminary Study. 2022; Available from: https://www.frontiersin.org/articles/10.3389/fpsyt.2022.763565/abstract.
- Xie, R.J., et al., A case report of Phelan-McDermid syndrome: preliminary results of the treatment with growth hormone therapy. Ital J Pediatr, 2021. 47(1): p. 49.
- Xie, R., et al., The Protective Role of E-64d in Hippocampal Excitotoxic Neuronal Injury Induced by Glutamate in HT22 Hippocampal Neuronal Cells. Neural Plast, 2021. 2021: p. 7174287.
- Li, T., et al., Clinical efficacy of atomoxetine combined with cognitive behavioral therapy in children with attention deficit hyperactivity disorder. Chinese Journal of Behavioral Medicine and Brain Science, 2021. 30(10): p. 916-922.
- Shellenberg, T.P., et al., An update on the clinical pharmacology of methylphenidate: therapeutic efficacy, abuse potential and future considerations. Expert Rev Clin Pharmacol, 2020. 13(8): p. 825-833.
- Caye, A., et al., Treatment strategies for ADHD: an evidence-based guide to select optimal treatment. Mol Psychiatry, 2019. 24(3): p. 390-408.
- De Sanctis, V., et al., Insulin-like Growth Factor-1 (IGF-1): Demographic, Clinical and Laboratory Data in 120 Consecutive Adult Patients with Thalassaemia Major. Mediterr J Hematol Infect Dis, 2014. 6(1): p. e2014074.
- Zheng, Y., et al., Reliability and validity of the Chinese version of Questionnaire - Children with Difficulties for Chinese children or adolescents with attention-deficit/hyperactivity disorder: a cross-sectional survey. Neuropsychiatr Dis Treat, 2018. 14: p. 2181-2190.
- Ke, X., et al., Risk factors for the difficulties in general activities across the day in Chinese children and adolescents with attention-deficit/hyperactivity disorder. Neuropsychiatr Dis Treat, 2019. 15: p. 157-166.
- Gordon, C.T. and S.P. Hinshaw, Parenting Stress as a Mediator Between Childhood ADHD and Early Adult Female Outcomes. J Clin Child Adolesc Psychol, 2017. 46(4): p. 588-599.
- Gili, M., et al., Comorbidity between common mental disorders and chronic somatic diseases in primary care patients. Gen Hosp Psychiatry, 2010. 32(3): p. 240-5.
- Liu, Q. and Y. Zhen, Interpretation of the second edition of the guidelines for the treatment of attention deficit hyperactivity disorder in china. Chinese Journal of Psychiatry, 2016. 49(3): p. 132-135.
- National Guideline, C., National Institute for Health and Care Excellence: Clinical Guidelines, in Attention deficit hyperactivity disorder: diagnosis and management. 2018, National Institute for Health and Care Excellence (UK)
Copyright © NICE 2018.: London.
Reviewer 2 Report
The study used QCD to retrospectively quantify the efficacy of atomoxetine treatment in the improvement of behavioral difficulties at different time and investigated the possible influence on the growth indicators in Chinese primary school children with ADHD. Overall the manuscript is well structured and the authors have establish the main outcome measure.
Introduction: please integrate the study into a theoretical framework model and add the recent studies focus on efficacy of the atomoxetine in school children with ADHD and the influence of this treatment on the growth indicators
Materials and methods: objectives of the study are not describing in this section; please clarify the design – is a retrospective or an open label study?
Results: a more detailed description on the assessment of side effect of medications it would be helpful,
Discussion: please add and compare the results with recent findings from studies focused on pharmacological treatament in ADHD (nonstimulants and stimulants drugs); discuss more the clinical implication of the results
Author Response
Q1:Introduction: please integrate the study into a theoretical framework model and add the recent studies focus on efficacy of the atomoxetine in school children with ADHD and the influence of this treatment on the growth indicators
Response: Based on the reviewer’s suggestion, we rewrote the introduction part and added this information: Previous studies applied IGF-1, IGFBP-3, and thyroid function tests as growth parameters to quantify the effects of methylphenidate on children's growth development [1-3]. However, few studies explored the impact of atomoxetine on children's growth development by these three growth parameters. Hence, this retrospective study was conducted to investigate the changes in serum IGF-1, IGFBP-3 levels, thyroid function tests, total protein (TP), albumin (ALB), and hemoglobin (HGB) to quantify the effects of atomoxetine on children's growth development in the short-term. Please see the highlighted parts in the revised manuscript
Q2: Materials and methods: objectives of the study are not describing in this section; please clarify the design – is a retrospective or an open label study?
Response: Thank you for your valuable suggestion. Based on your suggestion, we made the corrections and clarify the design is retrospective. We also added this information in materials and methods part: To quantify the effects of atomoxetine on different periods of a day and the efficacy to alleviate the core symptoms of ADHD. An inventory of the improvement in the children’s behavior and core symptoms of ADHD was kept using three parent-reported questionnaires electronically. To investigate the possible adverse reactions of atomoxetine, an adapted semi-structured interview was recorded every three weeks by the guardians or parents electronically. In addition, to quantify the effects of atomoxetine on children's growth development in the short-term, thyroid function, TP, ALP, and HGB levels were measured pre-treatment and post-treatment. The serum levels of IGF-1 and IGFBP-3 were also measured via the chemiluminescent immunometric assay (CLIA) method and analyzed in our hospital’s laboratory [4]. Please see the highlighted parts in the revised manuscript
Q3: Results: a more detailed description on the assessment of side effect of medications it would be helpful,
Response: Thank you for your valuable suggestion. Based on your suggestion, we rewrote this part and added this information: In this study, we focused on the most common drug adverse reactions such as severe adverse reactions (headache, insomnia, suicide attempt), gastrointestinal symptoms (loss of appetite, constipation, vomiting), fatigue, dizziness, and emotional instability based on the literature [5-7]. Please see the highlighted parts in the revised manuscript
Q4: Discussion: please add and compare the results with recent findings from studies focused on pharmacological treatament in ADHD (nonstimulants and stimulants drugs); discuss more the clinical implication of the results
Response: Thank you for your valuable suggestion. Based on your suggestion, we rewrote the discussion part and added this information: To decide first-line pharmacological treatment between methylphenidate and atomoxetine, the guardians or parents must be fully informed about the adverse reactions first [8]. This shared decision-making process needs physicians to cooperate with the guardians or parents. In China, the guardians or parents prefer to choose atomoxetine treatment for the following reasons: 1) adverse reactions on height; 2) they hope the effect of the drug last throughout the day; 3) the ADHD symptoms is not severe and immediate onset is not needed; 4) concerns about drug abuse [9-11] / Interestingly, previous studies indicated that parents found it more challenging to take care of ADHD children in the evening. They reported difficulties persuading their children to do their homework and enjoy family time [12, 13]. Alleviating the core symptoms of ADHD in the evening may effective solve this problem and reduce the parental stress level. Minimizing parenting stress and improving family interactions are important because they can reduce the risk of adverse outcomes in the long-term [13-15]. Consistent with our QCD results, atomoxetine treatment may be a choice in China to alleviate parenting stress and improve family interactions in children with ADHD. Please see the highlighted parts in the revised manuscript
- Bereket, A., et al., Height, weight, IGF-I, IGFBP-3 and thyroid functions in prepubertal children with attention deficit hyperactivity disorder: effect of methylphenidate treatment. Horm Res, 2005. 63(4): p. 159-64.
- Kim, W.J., et al., Preliminary Investigation of Association between Methylphenidate and Serum Growth Markers in Children with Attention-Deficit/Hyperactivity Disorder: A Cross-Sectional Case-Control Study. Soa Chongsonyon Chongsin Uihak, 2020. 31(3): p. 154-160.
- Toren, P., et al., Lack of effect of methylphenidate on serum growth hormone (GH), GH-binding protein, and insulin-like growth factor I. Clin Neuropharmacol, 1997. 20(3): p. 264-9.
- De Sanctis, V., et al., Insulin-like Growth Factor-1 (IGF-1): Demographic, Clinical and Laboratory Data in 120 Consecutive Adult Patients with Thalassaemia Major. Mediterr J Hematol Infect Dis, 2014. 6(1): p. e2014074.
- Pozzi, M., et al., Adverse drug events related to mood and emotion in paediatric patients treated for ADHD: A meta-analysis. J Affect Disord, 2018. 238: p. 161-178.
- Clavenna, A. and M. Bonati, Pediatric pharmacoepidemiology - safety and effectiveness of medicines for ADHD. Expert Opin Drug Saf, 2017. 16(12): p. 1335-1345.
- Ekhart, C., T. Vries, and F.V. Hunsel, Psychiatric adverse drug reactions in the paediatric population. Arch Dis Child, 2020. 105(8): p. 749-755.
- Wilens, T.E., et al., Clinically Meaningful Improvements in Early Morning and Late Afternoon/Evening Functional Impairment in Children with ADHD Treated with Delayed-Release and Extended-Release Methylphenidate. J Atten Disord, 2022. 26(5): p. 696-705.
- Shellenberg, T.P., et al., An update on the clinical pharmacology of methylphenidate: therapeutic efficacy, abuse potential and future considerations. Expert Rev Clin Pharmacol, 2020. 13(8): p. 825-833.
- Carucci, S., et al., Long term methylphenidate exposure and growth in children and adolescents with ADHD. A systematic review and meta-analysis. Neurosci Biobehav Rev, 2021. 120: p. 509-525.
- Caye, A., et al., Treatment strategies for ADHD: an evidence-based guide to select optimal treatment. Mol Psychiatry, 2019. 24(3): p. 390-408.
- Cortese, S., et al., Comparative efficacy and tolerability of medications for attention-deficit hyperactivity disorder in children, adolescents, and adults: a systematic review and network meta-analysis. Lancet Psychiatry, 2018. 5(9): p. 727-738.
- Gordon, C.T. and S.P. Hinshaw, Parenting Stress as a Mediator Between Childhood ADHD and Early Adult Female Outcomes. J Clin Child Adolesc Psychol, 2017. 46(4): p. 588-599.
- Hoza, B., Peer functioning in children with ADHD. Ambul Pediatr, 2007. 7(1 Suppl): p. 101-6.
- Muñoz-Silva, A., et al., Child/Adolescent's ADHD and Parenting Stress: The Mediating Role of Family Impact and Conduct Problems. Front Psychol, 2017. 8: p. 2252.

Round 2
Reviewer 1 Report
Thanks for giving me the opportunity of reviewing the revision of the manuscript.
I think the authors provided answers to my questions sufficiently most of time. However, there are still lingering issues to be further clarified for a few of the items. Please see my response below:
Q4 (page 2): It is still unclear the consenting was for the clinical treatment, or conduction of the study, or both. Did the authors obtain consent at the beginning of the treatment, with the idea of conducing research along with the clinical treatment? If so, it cannot be stated that this is a retrospective chart review study. I am not sure the relocation of hospital has anything to do with this issue either.
Q6 (Page 3): Again, it is still unclear if the collection of IGF-1/IGFBP-3 was a part of clinical work, or research, or both. If they were collected for clinical purpose, what was the guideline of how to interpret and use this information for clinical purpose? If they were just collected for research purpose, again it could not be stated as a retrospective chart review study, because clearly they were collected for research purpose from the beginning. Thus, this needs further clarification.
And there are still many grammar errors in the manuscript. For example, the newly added sentences in the abstract: “Although it is common knowledge that primary school children… in the evening. These two periods…studies of ADHD.” I think the authors meant to make these as one sentence so it has to be as “Although…in the evening, these two periods…studies of ADHD.” I highly recommend to thoroughly review and correct these type of errors before publication.
Author Response
Q1 (page 2): It is still unclear the consenting was for the clinical treatment, or conduction of the study, or both. Did the authors obtain consent at the beginning of the treatment, with the idea of conducing research along with the clinical treatment? If so, it cannot be stated that this is a retrospective chart review study. I am not sure the relocation of hospital has anything to do with this issue either.
Response: Thank you for your valuable suggestion and we were sorry for misunderstanding your meaning before. We thought your concern about the delayed approval date and it is due to the relocation of the hospital. The consent for reviewing the medical records to conduct this study was obtained from patients or guardians prior to the beginning of the treatment and was both for clinical treatment and the study. Based on your suggestion, we change our manuscript from retrospective to prospective throughout the manuscript. Please see the highlighted parts in the revised manuscript.
Q2 (Page 3): Again, it is still unclear if the collection of IGF-1/IGFBP-3 was a part of clinical work, or research, or both. If they were collected for clinical purpose, what was the guideline of how to interpret and use this information for clinical purpose? If they were just collected for research purpose, again it could not be stated as a retrospective chart review study, because clearly they were collected for research purpose from the beginning. Thus, this needs further clarification.
Response: Thank you for your valuable suggestion. IGF-1/IGFBP-3 was both a part of clinical work and research. Based on your suggestion we added this information in the discussion part: Short stature and growth deficits remain the most common pediatric concerns, and it is mainly presented with growth hormone (GH) deficiency accompanied by lower serum IGF-1 level [1, 2]. However, non-GH deficient short stature disorders such as idiopathic short stature (ISS) and growth hormone insensitivity (GHI) often presented with normal GH accompanied by lower serum IGF-1 level [3, 4]. Hence, the Growth Hormone Research Society (GRS) and Chinese official Guidelines for diagnosing and treating children with short stature recommended IGF-1 and its main binding protein IGFBP-3 as valuable biomarkers in assessing growth deficits[5, 6]. In addition, we change our manuscript from retrospective to prospective throughout the manuscript. Please see the highlighted parts in the revised manuscript.
Q3:And there are still many grammar errors in the manuscript. For example, the newly added sentences in the abstract: “Although it is common knowledge that primary school children… in the evening. These two periods…studies of ADHD.” I think the authors meant to make these as one sentence so it has to be as “Although…in the evening, these two periods…studies of ADHD.” I highly recommend to thoroughly review and correct these type of errors before publication.
Response: Thank you for your valuable suggestion. Based on your suggestion, this manuscript was carefully reviewed by an experienced editor who specializes in editing papers written by scientists whose native language is not English. The certificate of English editing would send to editor via email.
- Grimberg, A., et al., Guidelines for Growth Hormone and Insulin-Like Growth Factor-I Treatment in Children and Adolescents: Growth Hormone Deficiency, Idiopathic Short Stature, and Primary Insulin-Like Growth Factor-I Deficiency. Horm Res Paediatr, 2016. 86(6): p. 361-397.
- Collett-Solberg, P.F., et al., Growth hormone therapy in children; research and practice - A review. Growth Horm IGF Res, 2019. 44: p. 20-32.
- Anwar, G.M., et al., Study of primary IGF-1 deficiency in Egyptian children with idiopathic short stature. Horm Res Paediatr, 2013. 79(5): p. 277-82.
- Hwa, V., et al., Genetic causes of growth hormone insensitivity beyond GHR. Rev Endocr Metab Disord, 2021. 22(1): p. 43-58.
- Shen, Y. and D. Wang, Guidelines for the diagnosis and treatment of short children. Chinese Journal of Pediatrics, 2008(06): p. 428-430.
- Collett-Solberg, P.F., et al., Diagnosis, Genetics, and Therapy of Short Stature in Children: A Growth Hormone Research Society International Perspective. Horm Res Paediatr, 2019. 92(1): p. 1-14.